

# Predicting e-commerce product prices through the integration of variational mode decomposition and deep neural networks

Haojie Wu

Department of Business Administration, Shanxi Polytechnic College, Taiyuan, Shanxi, China

## ABSTRACT

Product prices frequently manifest nonlinear and nonstationary time-series attributes, indicating potential variations in their behavioral patterns over time. Conventional linear models may fall short in adequately capturing these intricate properties. In addressing this, the present study leverages the adaptive and non-recursive attributes of the Variational Mode Decomposition (VMD) methodology. It employs VMD to dissect the intricate time series into multiple Intrinsic Mode Functions (IMF). Subsequently, a method rooted in the minimum fuzzy entropy criterion is introduced for determining the optimal modal number (K) in the VMD decomposition process. This method effectively mitigates issues related to modal confusion and endpoint effects, thereby enhancing the decomposition efficacy of VMD. In the subsequent phase, deep neural networks (DNN) are harnessed to forecast the identified modes, with the cumulative modal predictions yielding the ultimate e-commerce product price prognostications. The predictive efficacy of the proposed Variational Mode Decomposition-deep neural network (VMD-DNN) decomposition model is assessed on three public datasets, wherein the mean absolute percentage error (MAPE) on the E-commerce Price Prediction Dataset and Online Retail Dataset is notably low at 0.6578 and 0.5414, respectively. This corresponds to a remarkable error reduction rate of 66.5% and 70.4%. Moreover, the VMD-DNN decomposition model excels in predicting e-commerce product prices through DNN, thereby amplifying the VMD decomposition capability by 4%. The VMD-DNN model attains superior results in terms of directional symmetry, boasting the highest Directional Symmetry (DS) score of 86.25. Notably, the forecasted trends across diverse price ranges closely mirror the actual trends.

# INTRODUCTION

In the era of highly evolved e-commerce, a diverse array of online data is available. Even in the absence of historical interaction data for new goods, descriptive information such as images is typically present. Conversely, mature commodities often feature substantial user evaluation text. Both image and text data play pivotal roles in e-commerce, constituting rich sources of information. However, conventional price prediction methods encounter

Corresponding author
Haojie Wu, 15234097680@163.com

challenges in extracting valuable insights from images and text, thereby failing to offer adequate support for comprehensive price prediction systems. In the current landscape of a complex and volatile market economy, precise product price prediction assumes paramount importance for companies in formulating strategic decisions. Traditional price forecasting approaches confront multiple challenges stemming from market complexity, nonlinearity, and nonstationary time series properties (*Cortez et al., 2018*; *Wang et al., 2020*). A fundamental limitation arises from the inherent difficulty of traditional linear models in capturing the diverse behavioral patterns exhibited by product prices over time.

Commodity price prediction represents a vital and intricate endeavor within the realm of e-commerce. The dynamic life cycle of commodities introduces price fluctuations (*Agnello et al., 2020*). New or nascent commodities may grapple with data sparsity in price prediction due to the absence of historical interaction data, while mature commodities often possess extensive historical price data. Traditional price prediction algorithms may falter in addressing these disparities. Current price forecasting systems typically overlook the influence of the commodity life cycle, adhering to a singular algorithmic strategy. However, a singular algorithm may not aptly accommodate the nuances of price prediction for new and mature commodities, potentially undermining prediction performance (*Lago et al., 2021*). Consequently,it is advisable to establish a comprehensive price forecasting framework that integrates multiple advanced forecasting algorithms. This framework should discern the commodity's life cycle stage based on its characteristics and historical price data, thereby allowing for the targeted adoption of suitable algorithms to enhance the accuracy of price forecasting.

At the present juncture, the integration of unstructured data into price prediction primarily involves the extraction of sentiment features, keyword features, or event features from media texts. This entails simultaneous prediction with futures prices. The crux of pertinent research lies in the proficient conversion of unstructured data and the adept identification and assimilation of valid information into e-commerce product prediction (*Pan & Zhou, 2020*; *Ramkumar et al., 2023*; *Sun et al., 2022*). The following issues merit exploration: firstly, the presence of potentially redundant information, such as analytical reports and social comments, introduces noise into the extracted effective price features, potentially influencing the model's discernment of price fluctuations; secondly, existing methodologies extracting event features from text data necessitate extensive manual annotations on specific corpora, susceptible to subjective judgment, and may inadvertently disregard other pertinent information in the text; thirdly, the fusion of structured price data and unstructured textual information features remains a subject of debate. Some studies directly concatenate trading data, substantial financial indicator data, and a solitary sentiment feature as inputs to the prediction model, without accounting for disparities in data features and dimensions.

In the domain of complex time series predictive modeling, the decomposition integration methodology is hailed as an effective strategy to enhance prediction accuracy. Its fundamental concept involves utilizing signal decomposition algorithms to break down a complex time series into a series of relatively simple and smooth subsequences, thereby reducing the complexity of modeling efforts (*Da Silva RG et al., 2020*). The Variational

Mode Decomposition (VMD), a signal processing technique, adeptly decomposes complex signals into multiple eigenmodes, facilitating a nuanced understanding of implicit modes in the data. Concurrently, the deep neural network (DNN) processes unstructured data through its intricate structure, capturing intricate relationships in price changes and enhancing prediction accuracy by diversifying input data (*Güvenç, Çetin & Koçak, 2021*). DNN excels in feature extraction from image and text coding, facilitating the discernment of nonlinear relationships and adapting more effectively to dynamic changes in e-commerce product prices. The combination of VMD and DNN emerges as an innovative solution for e-commerce product price prediction.

The primary goal of this article is to develop a comprehensive and robust framework for predicting e-commerce product prices by leveraging advanced forecasting algorithms that integrate both structured and unstructured data. The focus is on enhancing prediction accuracy by addressing the limitations of traditional price prediction methods, especially in the context of nonlinear and nonstationary time-series data and the dynamic life cycles of commodities.

The main tasks include:

1. Advanced decomposition technique using VMD and minimum fuzzy entropy criterion: The study leverages the adaptive and non-recursive attributes of VMD to dissect intricate time-series data into multiple IMFs. It introduces a method based on the minimum fuzzy entropy criterion to determine the optimal number of modes (K) in the VMD decomposition process, effectively mitigating issues related to modal confusion and endpoint effects.

2. Integration of deep neural networks for mode forecasting: The study employs DNN to forecast the identified modes from the VMD decomposition. This integration allows for accurate prediction of e-commerce product prices by combining the strengths of VMD in decomposition and DNN in predictive modeling, thereby enhancing the overall efficacy of the forecasting method.

3. Comprehensive methodology for time-series analysis: By combining VMD with a minimum fuzzy entropy criterion and DNN, the study presents a comprehensive and innovative methodology for analyzing and forecasting nonlinear and nonstationary time-series data. This approach addresses the limitations of conventional linear models, providing a robust framework for improved time-series analysis.

## LITERATURE REVIEW

The neural network algorithm, adept at handling intricate relationships, proves highly suitable for e-commerce data prediction. Possessing characteristics of substantial parallelism and pronounced nonlinearity, neural network models with diverse structures find swift application among scholars in modeling nonlinear systems, showcasing notable applicability in time series analysis. *Pandey et al. (2022)* employed ARIMA and a radial basis function-based neural network to predict exchange rates, validating the enhancement of prediction capabilities over linear models. *Li et al. (2021)* achieved a significant reduction in prediction error by employing a neural network with fewer nodes in the hidden layer and

a lower termination condition, realizing an approximate 60% improvement. *Deebak & Al-Turjman (2022)* introduced a five-layer DNN model, proving more effective in predicting the rise and fall of a given product. Recognizing the limitations of DNN in accurately predicting time series variations, the advent of recurrent neural networks resolves this issue. *Issaoui et al. (2021)* demonstrated effective e-commerce market trend prediction using the long short-term memory (LSTM) method. Given that real-world time series seldom adhere strictly to linearity or nonlinearity, hybrid prediction methods yield more satisfactory results. Numerous studies (*Bukhari et al., 2020*; *Niu, Xu & Wang, 2020*; *Zhang & Chen, 2023*) showcase improved prediction accuracy by combining different methods and models compared to employing a singular model. Studies have demonstrated that DNN can effectively capture intricate trends and seasonal fluctuations in time series data, offering significant advantages over traditional methods (*Tan et al., 2023*). Furthermore, DNN are capable of constructing multi-level prediction models by integrating diverse input features, such as macroeconomic indicators, market sentiment data, and technical indicators, thereby enhancing predictive accuracy (*Sharma & Mehta, 2024*). Moreover, researchers have explored combining DNNs with other machine learning algorithms to boost price forecasting performance. For instance, *Nanjappa et al. (2024)* proposed a method that integrates LSTM networks with DNN for stock price prediction. Their results indicated that this hybrid model excels in capturing both short-term and long-term dependencies. Other studies have investigated DNN models incorporating attention mechanisms to better balance the importance of input features and improve prediction accuracy (*Li et al., 2024*; *Hu et al., 2024*).

Owing to the non-stationary and non-linear attributes inherent in e-commerce product prices, the significance of data preprocessing cannot be overstated. *Gu (2023)* undertook the prediction of e-commerce product prices, utilizing the Complementary Integrated Empirical Modal Decomposition model tailored for nonlinear, complex, and irregularly distributed data. In a similar vein, *Osama et al. (2023)* engineered a hybrid decomposition-forecasting model adept at capturing the nonlinearities and volatilities intrinsic to time-series characteristics. Robustness tests were conducted to ensure data integrity, effectively addressing modal mixing issues, significantly reducing data reconstruction errors, and fitting nonlinear data. To enhance prediction accuracy, various reconstruction methods are employed in the post-decomposition reconstruction process for both linear and nonlinear data. Typically, linear integration involves a straightforward summation of model predictions. However, this approach lacks a robust foundation and proves unsuitable for nonlinear data, such as e-commerce product prices. As a remedy, intelligent models are now widely employed for the nonlinear reconstruction of sequences.

Empirical Mode Decomposition (EMD), rooted in the concept that signals can adaptively generate intrinsic mode functions, is extensively employed for recursively decomposing signals into distinct yet unknown and independent modes (*Campi, 2022*). Recognizing the limitations of EMD concerning noise and sampling sensitivity, scholars have explored alternative approaches. *Dragomiretskiy & Zosso (2013)* introduced Variational Modal Decomposition, which demonstrates enhanced robustness to sampling and noise, offering an effective solution to signal decomposition challenges and providing a novel avenue

for further research. *Xu & Ren (2019)* applied the VMD algorithm to decompose chaotic time series, resulting in significantly improved prediction performance. *Guo et al. (2022)* proposed the integration of the VMD algorithm and a generalized neural network for predicting chaotic time series, with simulation results highlighting superior accuracy for the VMD-GRNN model compared to the EMD-GRNN model. *Jiang, Han & Wang (2020)* devised a stacked recurrent neural network model based on the VMD algorithm, demonstrating excellent performance in long-term prediction.

Despite the demonstrated effectiveness of VMD decomposition, its outcomes are contingent on the selection of the key parameter modal number (K value). An inappropriate choice, either excessive or insufficient decomposition, can compromise the accuracy of analysis results (*He et al., 2021*). Consequently, selecting an appropriate K value before decomposition emerges as a pivotal factor for the widespread application of VMD. Addressing the challenge of K value selection, *Zhang et al. (2020)* employed VMD to decompose wind speed time series, optimizing the decomposition K value through Sample Entropy calculations. However, sample entropy, utilizing a binary function for similarity measurement, may yield inaccurate or undefined results. Fuzzy entropy, reflecting the complexity of time series, exhibits an increase in entropy value when disturbances in the time series lead to heightened uncertainty in state values (*Wang et al., 2023*). Furthermore, DNN showcase formidable computational capabilities, particularly when handling large datasets, demonstrating self-learning, self-adaptation, and the ability to approximate complex nonlinear relationships comprehensively, surpassing other machine learning algorithms.

## METHODOLOGY

In this study, fuzzy entropy is incorporated to optimize VMD for determining the appropriate modal number K, building upon the foundation of sample entropy. In contrast to the binary function employed in sample entropy, the utilization of fuzzy affiliation functions for similarity measurement, along with fuzzy boundary measurements, enhances the evaluation of signal complexity. This approach results in more continuous and smoother changes in entropy values. Following the normalization of data through the VMD method, the decomposed modal components are subjected to prediction *via* DNN. The Variational Mode Decomposition-deep neural network (VMD-DNN) fusion model generates predicted values by aggregating the individual prediction outcomes.

### Moving window structure

As shown in Fig. 1, the prediction structure employed in this article adopts a moving window format, where predictions within each window are independent and do not mutually influence one another. This logical framework serves a dual purpose: firstly, it simulates real trading scenarios, mitigating the occurrence of ex post prediction errors. Secondly, it satisfies the condition of parallelism across different time windows or within different IMFs within each window, thereby expediting the model training process (*Nava,*

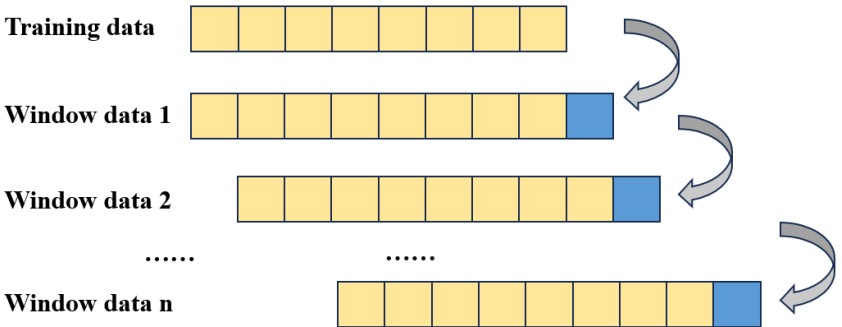

**Figure 1  Windowed data structure.**

*Di Matteo & Aste, 2018*). Within the sliding window framework, each window's data corresponds to the original time series.

### Optimization of VMD based on fuzzy entropy

The variational problem is initially formulated by assuming the decomposition of the original signal f into k components. It is imperative to ensure that the decomposed sequence represents modal components with center frequency and finite bandwidth. Simultaneously, the task involves determining the estimated bandwidth of each modal sum to be the smallest. Additionally, the summation of all modes is required to be equal to the original signal, serving as a constraint. Subsequently, the corresponding constraint variational expression is established.

$$\min_{\{u_k\},\{\omega_k\}} \left\{ \sum_k \left\| \partial_t \left[ \left( \delta(t) + j/\pi t \right) * u_k(t) \right] e^{-j\omega_k t} \right\|_2^2 \right\} \quad \text{s.t.} \quad \sum_{k=1}^{K} u_k = f \tag{1}$$

where $K$ is the number of modes (positive integers) to be decomposed, and $\{u_k\}, \{\omega_k\}$ are the first $k$ modal component and the corresponding center frequency of the decomposed modes, where $\delta(t)$ is the Dirac function, and $*$ is the convolution operator.

To optimize the modal number in VMD decomposition, fuzzy entropy is employed in this study. The process of calculating fuzzy entropy for optimizing the VMD decomposition model is as follows:

Reconstruct the phase space of the original sequence to obtain a m vector

$$\left\{ x_i^m \right\} (i = 1, 2, \ldots, N - m + 1). \tag{2}$$

Then,

$$\left\{ x_i^m \right\} = \left\{ x(i+j) - x_0(i) \right\}, j = 0, 1, \ldots, m - 1 \tag{3}$$

and,

$$x_0(i) = \frac{1}{m} \times \sum_{j=0}^{m-1} x(i+j). \tag{4}$$

Calculate the distance between time series $x_i^m$ and $x_j^m$.

$$d_{i,j}^m = \max_{k \in (0,m-1)} \left| (x(i+k) - x_0(i)) - (x(j+k) - x_0(j)) \right|. \tag{5}$$

For a given n and r , use the fuzziness function $u\left(d_{i,j}^m, n, r\right)$. Calculate the time series $x_i^m$ with $x_j^m$ the similarity of the time series with $D_{i,j}^m(n,r) = u\left(d_{i,j}^m, n, r\right)$.

For the time series $x_i^m$ , define the following function.

$$\phi^m(n,r) = \frac{1}{N-m} \sum_{i=1}^{N-m} \left( \frac{1}{N-m-1} \sum_{j=1, j \neq i}^{N-m} D_{i,j}^m \right). \tag{6}$$

Define the fuzzy entropy:

$$FE(m,n,r,N) = \ln\phi^m(n,r) - \ln\phi^{m+1}(n,r). \tag{7}$$

The calculation of fuzzy entropy is associated with the parameters m, n, and r. The embedding dimension (m) is typically set to 2, as it is computationally less intensive and more responsive to sequence changes. The similarity tolerance limit (r) is chosen as 0.2 times the standard deviation of the sequence (std), ensuring that r is not excessively large. The standard deviation of the sequence (std) is usually taken as the value of n, and n is commonly set to 1.

Since fuzzy entropy can measure the complexity of time series, based on this, optimization based on the minimum fuzzy entropy criterion VMD decomposition of modal K The specific steps of the optimization method are as follows: Firstly, given that $K = 3, 4, \cdots, 14$VMD model to decompose the original time series $x(t)(t = 1, 2, \cdots, N)$ model to adaptively decompose the original time series into K different scales of the IMF set of component sequences $\{I_i(t)\}(i = 1, 2 \cdots, K)$. Then, the IMF components are calculated and ranked by the fuzzy entropy $\{I_i(t)\}$. The IMF components exhibiting the least fuzzy entropy are designated as trend terms, while the remaining IMF components are denoted as stochastic disturbance terms. Subsequently, a comparative analysis of the fuzzy entropy associated with the trend term is conducted across varying decomposition values (K values) to ascertain the optimal number of decompositions. As the K value decreases, the fuzzy entropy of the trend term escalates, and conversely, with an increase in the K value, the fuzzy entropy of the trend term expands. However, with a continued rise in the K value, the fuzzy entropy demonstrates a tendency to gradually stabilize. Consequently, to prevent excessive decomposition, the inflection point, where the fuzzy entropy begins to stabilize, is identified as the modal number for the VMD decomposition.

## DNN structure

The network architecture of the DNN is depicted in Fig. 2, encompassing multiple hidden layers, an input layer, and an output layer (*Aldahdooh et al., 2022*). The input layer is denoted as $X = [x_1, x_2, x_n]^7$. The input data comprises a column vector denoted as n-dimensional. The dataset encompasses seven distinct types of information:

Daily product prices: Detailed price records for each day of the product.

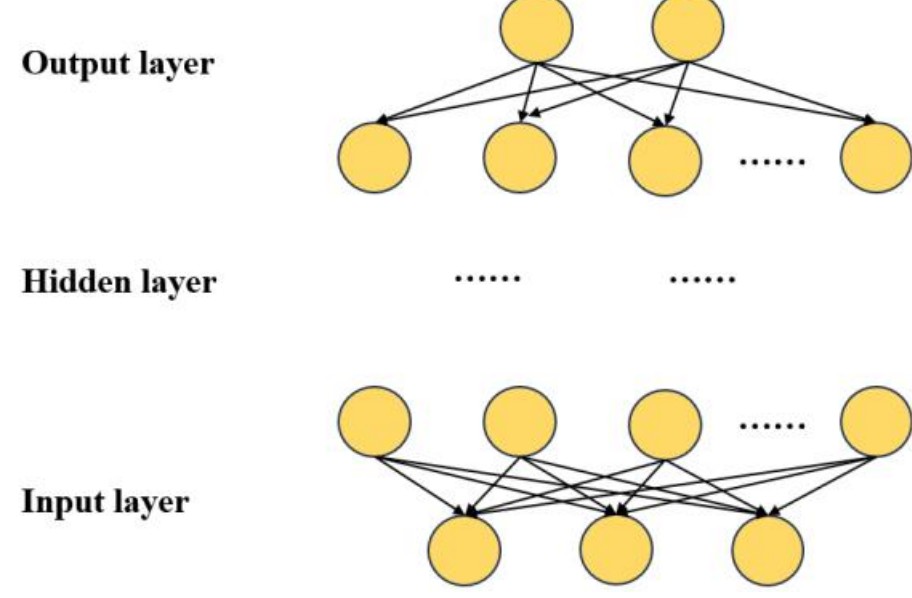

**Figure 2** DNN network structure.

Product dimensions: Numeric representation of product dimensions, including length, width, and height.

Seasonal sales volume: Quantification of the impact of seasonality on product sales volume, usually on a monthly or quarterly basis.

Competitor prices: Specific price details for comparable products offered by competitors.

User ratings: User-assigned ratings for a given product.

Promotion discount percentage: The specific percentage of product discount during promotional activities.

Inflation rate: The inflation rate within the economic environment, expressed as a percentage.

Once categorized and processed as output values, this data is transmitted from the input layer to the hidden layer, facilitating the establishment of the initial relationship between hidden inputs and outputs, as represented by Eq. (8).

$$R_1 = f(w_1 \cdot X + b_1) \tag{8}$$

where $R_1$ represents the output matrix of the first hidden layer, while $W_i$ and $b_i$ stand for the weight parameter and threshold parameter between the input layer and the hidden layer.

If we designate the variable for the p-th element of the first hidden layer as $r_{1,p}$, where p is the index, then $w_{1,p}$ represents the weight matrix between the input layer and the first hidden layer, and $b_{1,p}$ represents the value corresponding to the first variable in the vector of values between the input layer and the first hidden layer. Each output value in

**Table 1  DNN structure settings.**

| Layer nuber | Layer name | Layer type | Output dimension | Value |
|---|---|---|---|---|
| 1 | dense_1 | Dense | (None, 128) | 896 |
| 2 | activation_1 | Activation | (None, 128) | 0 |
| 3 | dropout_1 | Dropout | (None, 128) | 0 |
| 4 | dense_2 | Dense | (None, 64) | 8,256 |
| 5 | activation_2 | Activation | (None, 64) | 0 |
| 6 | dropout_2 | Dropout | (None, 64) | 0 |
| 7 | dense_3 | Dense | (None, 3) | 195 |

the activation function is determined accordingly.

$$r_{1,p} = F\left( \sum_{i=1}^{n} w_{1,p_i} \cdot X_i + b_{1,p} \right). \tag{9}$$

According to the principle of DNN, the output of the previous hidden layer is the input of the next hidden layer, so the output of the first hidden layer of the DNN model is the input of the next hidden layer. m The output of the first hidden layer of the DNN model $R_m$ of the first hidden layer of the DNN model is given by

$$R_m = f(w_m \cdot R_{m-1} + b_m). \tag{10}$$

Input X after undergoing processing by the input layer, is forwarded to the hidden layer. Following the completion of processing in the hidden layer, the result is transmitted to the output layer. This process can be expressed as follows.

$$y = g(W_{n+1} \cdot R_n + b_{n+1}). \tag{11}$$

To enable the DNN for predicting the price movements of e-commerce products—up, down, or flat—the network structure is designed as outlined in Table 1.

The architectural design of the DNN takes into consideration the functions and parameters of different layers. The entire network comprises seven layers, including the input layer, two hidden layers, a dropout loss layer, and the output layer.

Commencing with the input layer, responsible for receiving raw data, the information proceeds to the first hidden layer. This layer encompasses 64 neurons, each associated with a feature from the input layer, with the objective of extracting key features from the input data. Subsequently, the second hidden layer, more intricate than the first, with 128 neurons, learns an abstract representation of the data at a deeper level. To prevent overfitting during training, a dropout loss layer is introduced between the two hidden layers. The dropout layer mitigates overfitting by randomly deactivating a number of neurons with a specific probability during training.

Ultimately, the output layer employs a sigmoid function as an activation function to determine the probability of belonging to the three categories. The sigmoid function maps output values between 0 and 1, representing the probability of belonging to each category, facilitating multi-category classification tasks.

To optimize training, adaptive learning rate methods and learning rate schedules are employed. Specifically, the Adam optimizer is used for its capability to adjust the learning rate dynamically based on the estimates of first and second moments of gradients. The Adam optimizer updates the weights as follows:

$$m_t = \beta_1 m_{t-1} + (1 - \beta_1) \nabla \mathfrak{L}_t$$
$$v_t = \beta_2 v_{t-1} + (1 - \beta_2)(\nabla \mathfrak{L}_t)^2$$
$$\hat{m}_t = \frac{m_t}{1 - \beta_1^L}$$
$$\hat{v}_t = \frac{v_t}{1 - \beta_2^L}$$
$$W_t = W_{t-1} - \frac{\alpha \hat{m}_t}{\sqrt{\hat{v}_t} + \epsilon}$$

where $m_t$ and $v_t$ are the first and second moment estimates, respectively, $\beta_1$ and $\beta_2$ are decay rates for the moments, $\alpha$ is the learning rate, and $\epsilon$ is a small constant to prevent division by zero.

Additionally, learning rate schedules are implemented to adjust the learning rate during training. For example, a step decay schedule reduces the learning rate by a factor after a set number of epochs, allowing the model to converge more effectively over time.

## VMD-DNN

The block diagram illustrating the structure of the VMD-DNN price prediction model constructed in this article is depicted in Fig. 3.

The specific modeling steps are as follows:

Step 1: Optimization of K value. Calculate the fuzzy entropy of the trend term under various VMD decomposition modes and identify the optimal parameter values for decomposition based on the minimum fuzzy entropy criterion (K value).

Step 2: Time series decomposition. Following the determination of the optimal K value, employ the VMD model to adaptively decompose the original time series into a set of K IMF component sequences $\{I_i(t)\}(i = 1, 2, \ldots, K)$.

Step 3: DNN input. Utilize the trained DNN model for each eigenmode function as input and obtain the corresponding prediction results.

Step 4: Integration prediction. Employ the ELM model to train and predict the trend and random interference terms within the IMF component. This process results in the predicted values of the IMF component $s_i = \{s_1, s_2, s_3, \ldots, s_K\}$, then linearly sum the predicted values of IMF components $s_i$ to derive the final prediction results.

The pseudo-code on which the VMD-DNN model runs is shown in Algorithm 1.

## EXPERIMENTATION AND ANALYSIS

The proposed VMD-DNN model is designed for application in the e-commerce product price prediction task. The model is trained using three datasets, and its performance is compared with existing approaches, namely ELM (*Weng et al., 2020*), DNN (*Güvenç, Çetin & Koçak, 2021*), VMD-ELM (*Dabin, Liling & Liwen, 2023*), and VMD-SVR (*Liu et al.,*

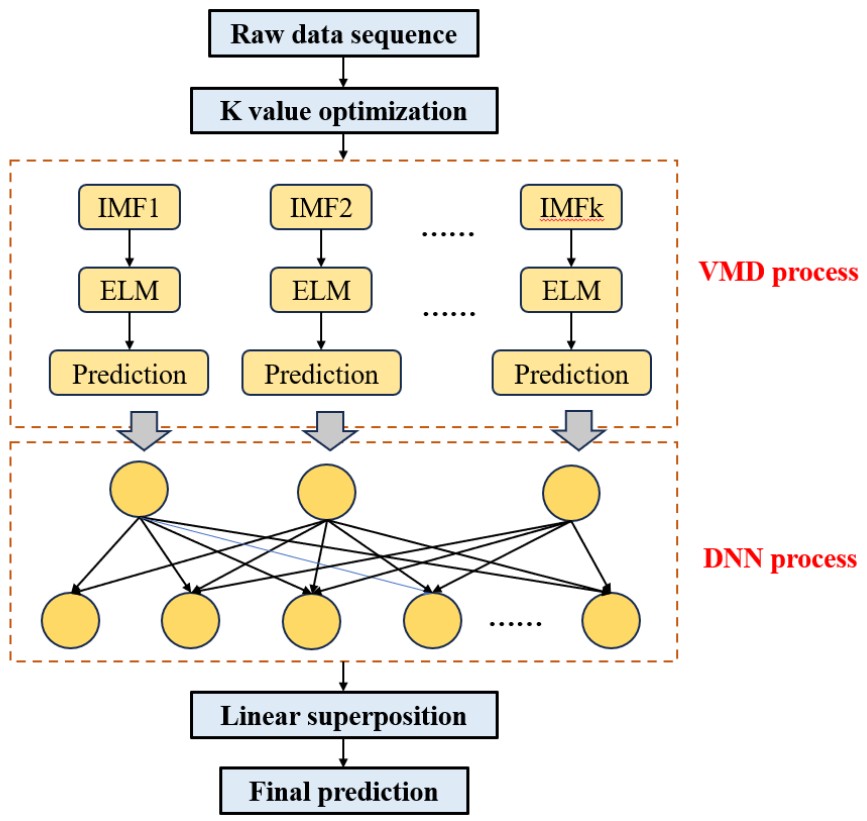

**Figure 3  Structure of the VMD-DNN.**

*2021*). ELM is recognized for its rapid training and strong generalization capabilities, while DNN is well-suited for intricate pattern learning in large-scale, high-dimensional datasets. VMD-ELM and VMD-SVR integrate VMD with ELM and SVR, respectively, leveraging signal decomposition to extract eigenmodes for enhanced data characterization.

## Data sets

The utilized datasets for model training encompass three publicly available sources, contributing diverse and comprehensive information to enhance the model's comprehension of the intricate relationship between products and prices.

Amazon Product Reviews Dataset (DOI 10.5281/zenodo.6657410): Derived from Amazon product reviews, this dataset includes extensive product information, user reviews, and ratings. It offers a substantial amount of data, potentially containing details pertinent to product prices. The inclusion of multiple data layers aids the model in capturing correlations between user feedback and pricing dynamics.

E-commerce Price Prediction Dataset (DOI 10.5281/zenodo.11237099): Sourced from Kaggle, a platform for open data science competitions, this dataset is tailored for e-commerce product price prediction. Encompassing various product categories and related features, it provides the model with diverse training instances, fostering experience in predicting prices within different contextual settings.

Algorithm 1: VMD-DNN main program

**Input:**

**original_time_series: Time series data to be analyzed**

**min_K: Minimum value of K to be considered**

**max_K: Maximum value of K to be considered**

**Output:**

**final_prediction: Integrated prediction result**

**1. Function Optimize_K(original_time_series, min_K, max_K)**

**Initialize best_K to None**

**Initialize min_fuzzy_entropy to infinity**

**For K from min_K to max_K inclusive**

**fuzzy_entropy = CalculateFuzzyEntropy(original_time_series, K)**

**If fuzzy_entropy < min_fuzzy_entropy then**

**min_fuzzy_entropy = fuzzy_entropy**

**best_K = K**

**Return best_K**

**2. Function Perform_VMD_Decomposition(original_time_series, optimal_K)**

**IMF_components = VMD_Decompose(original_time_series, optimal_K)**

**Return IMF_components**

**3. Function Apply_DNN_Model(IMF_components)**

**Initialize predictions as an empty list**

**For each IMF_component in IMF_components**

**predicted_values = DNN_Predict(IMF_component)**

**Append predicted_values to predictions**

**Return predictions**

**4. Function Integrate_Predictions(predictions)**

**trained_ELM_model = Train_ELM_Model(predictions)**

**IMF_predictions = ELM_Predict(trained_ELM_model, predictions)**

**final_prediction = Linear_Combination(IMF_predictions)**

**Return final_prediction**

**// Execution**

**optimal_K = Optimize_K(original_time_series, min_K, max_K)**

**IMF_components = Perform_VMD_Decomposition(original_time_series, optimal_K)**

**predictions = Apply_DNN_Model(IMF_components)**

**final_prediction = Integrate_Predictions(predictions)**

Online Retail Dataset (DOI 10.24432/C5BW33): Encompassing online retail information, this dataset comprises sales data, user behavior patterns, and price variations across diverse products. Analysis of this dataset facilitates the model's understanding of price fluctuations under various conditions, thereby enhancing its proficiency in accurately predicting e-commerce product prices.

**Table 2  Model performance under different experimental parameters.**

| Batch size | Learning rate | Activation function | Accuracy | Loss |
|---|---|---|---|---|
| 16 | 0.001 | ReLU | 87.5% | 0.25 |
| 16 | 0.001 | Tanh | 86.8% | 0.28 |
| 16 | 0.001 | Sigmoid | 85.0% | 0.30 |
| 16 | 0.01 | ReLU | 89.2% | 0.20 |
| 16 | 0.01 | Tanh | 88.5% | 0.23 |
| 16 | 0.01 | Sigmoid | 86.3% | 0.27 |
| 32 | 0.001 | ReLU | 85.6% | 0.30 |
| 32 | 0.001 | Tanh | 84.9% | 0.32 |
| 32 | 0.001 | Sigmoid | 83.5% | 0.35 |
| 32 | 0.01 | ReLU | 87.8% | 0.25 |
| 32 | 0.01 | Tanh | 86.2% | 0.28 |
| 32 | 0.01 | Sigmoid | 84.8% | 0.32 |
| 64 | 0.001 | ReLU | 82.3% | 0.35 |
| 64 | 0.001 | Tanh | 81.5% | 0.38 |
| 64 | 0.001 | Sigmoid | 80.9% | 0.40 |
| 64 | 0.01 | ReLU | 84.0% | 0.30 |
| 64 | 0.01 | Tanh | 82.7% | 0.32 |
| 64 | 0.01 | Sigmoid | 82.1% | 0.37 |

## Experiments details and evaluation indicators

This article uses the CPU of Xeon[(R)] E5-2640 v4, the GPU of 4*Nvidia Tesla V100 and the ubuntu system to complete the environment setup and model training. The deep learning framework is Tensorflow. Table 2 presents the experimental results for different batch size, learning rate, and activation function.

The analysis underscores that ReLU, with a batch size of 16 and a learning rate of 0.01, represents the most effective combination for achieving optimal model performance. This configuration not only delivers the highest accuracy but also minimizes the loss, thereby demonstrating superior robustness and stability. Although Tanh offers competitive performance, it does not reach the same level of effectiveness as ReLU. Sigmoid, while still useful in certain contexts, shows comparatively lower performance metrics. Consequently, ReLU is recommended as the preferred activation function for models requiring high accuracy and low loss, based on the results from this experimental evaluation.

To comprehensively assess the performance of various prediction models, this article employs root mean square error (RMSE), mean absolute percentage error (MAPE), mean absolute error (MAE), and direction symmetry (DS) as evaluation criteria (*Ardiansyah, Majid & Zain, 2016*). In this study, RMSE, MAPE, MAE, and DS serve as the evaluation metrics (*Ardiansyah, Majid & Zain, 2016*), with a lower RMSE, MAPE, and MAE indicative of higher prediction accuracy. Conversely, a higher value of DS signifies a better alignment with the actual data trend.

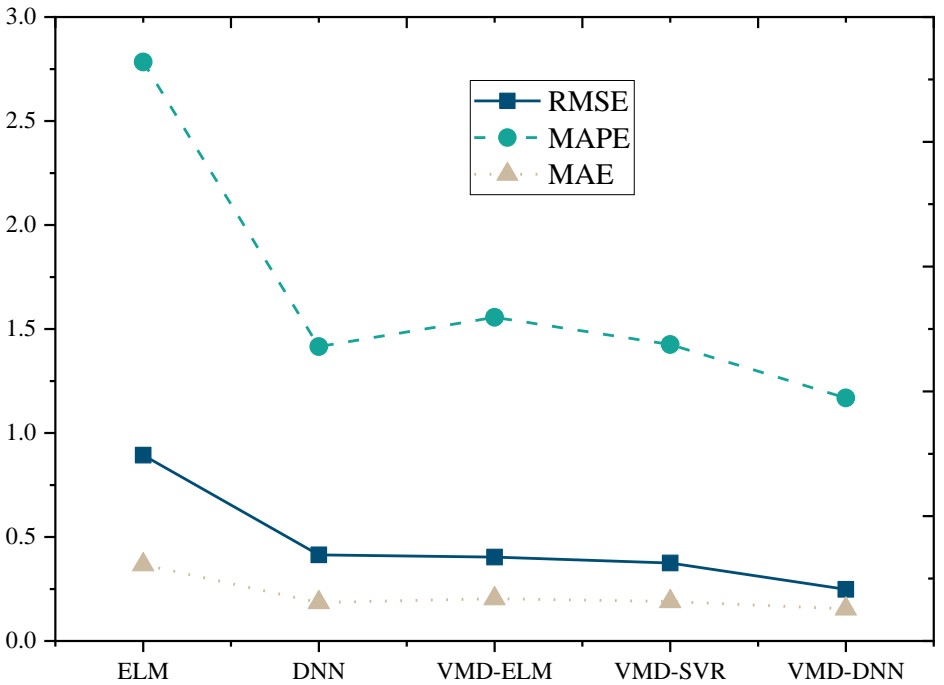

**Figure 4    Model comparison under Amazon product reviews dataset.**

## Results and Discussion

The performance comparison of models on the Amazon Product Reviews Dataset (Fig. 4) reveals that, in terms of RMSE, the VMD-DNN model outperforms others, achieving the lowest error value (0.2481). This superiority in accuracy positions the VMD-DNN model as a clear leader among the models. Furthermore, in terms of MAPE and MAE, the VMD-DNN model demonstrates the best performance, with values of 1.1686 and 0.1546, respectively, indicating closer proximity to true values compared to other models.

The DNN and VMD-SVR models also exhibit commendable performance, showcasing effective error reduction compared to the traditional ELM and VMD-ELM models. Conversely, the ELM model performs relatively poorly across all metrics, potentially due to its limitations in nonlinear modeling. In summary, the VMD-DNN model excels in e-commerce product price prediction, offering a robust solution to enhance prediction accuracy and practical utility.

Furthermore, upon comparing the single prediction model with the decomposition integration prediction model (Figs. 5 and 6), it is evident that all three error metrics and directional metrics of the decomposition integration prediction models surpass those of the single model. This observation underscores the effectiveness of the decomposition integration strategy.

Notably, the proposed VMD-DNN sub-model achieves an impressive MAPE of only 0.6578 on the E-commerce Price Prediction Dataset and 0.5414 on the Online Retail Dataset. The associated error reduction rates of 66.5% and 70.4%, respectively,

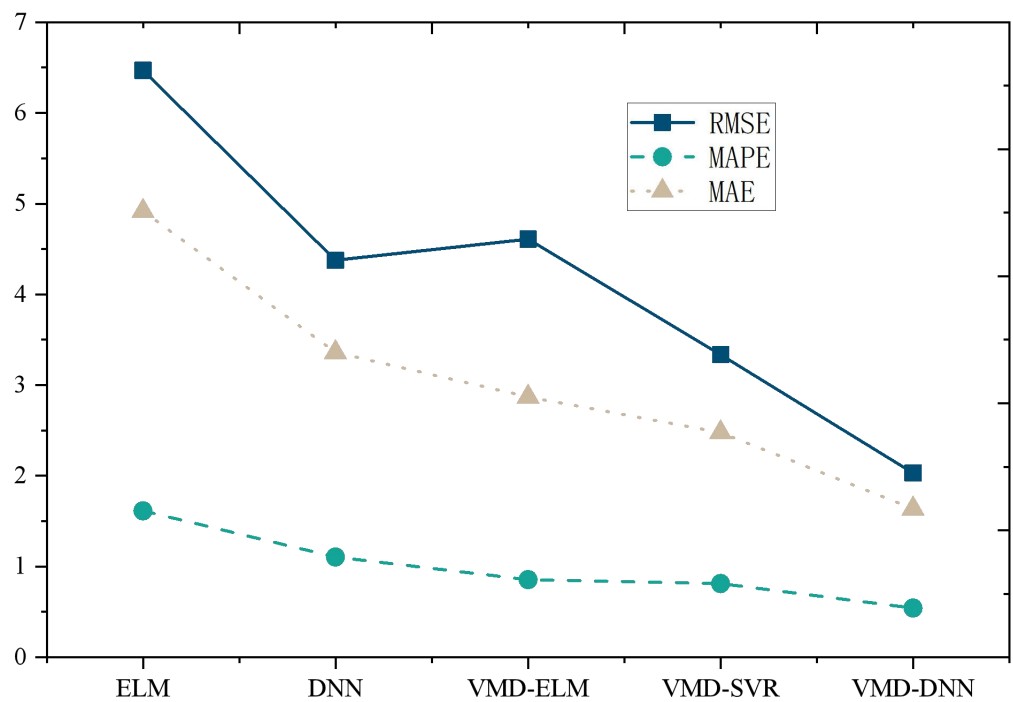

**Figure 5** Model comparison under E-commerce price prediction dataset.

compared to the single model, emphasize the substantial improvement achieved through the decomposition integration approach. Relative to other decomposition integration models, VMD-DNN excels in breaking down complex e-commerce product data into more manageable sub-sequences, alleviating the prediction burden on the single model. This enhancement augments its generalization capability, ultimately leading to optimal prediction performance.

DS provides a more nuanced analysis of model performance across various datasets, as illustrated in Fig. 7. This metric offers insight into the consistency of model predictions in capturing directional trends, complementing traditional accuracy measures.

In the Amazon Product Reviews Dataset, the VMD-DNN model achieves a notable DS score of 86.25, highlighting its excellence in both prediction accuracy and directional alignment. This high DS score indicates that the VMD-DNN model not only performs well in terms of accuracy but also effectively captures the underlying trend directions. The VMD-ELM model also demonstrates significant performance with a DS score of 79.3, showcasing its ability to maintain good directional symmetry, albeit slightly behind the VMD-DNN model.

The exceptional performance of the VMD-DNN model is further confirmed in the E-commerce Price Prediction Dataset and the Online Retail Dataset, where it secures the highest DS scores of 87.5 and shows consistent performance across datasets. This consistency underscores the model's robustness in maintaining both prediction accuracy and directional alignment. The VMD-ELM model, with DS scores of 79.8 and 79.3 in
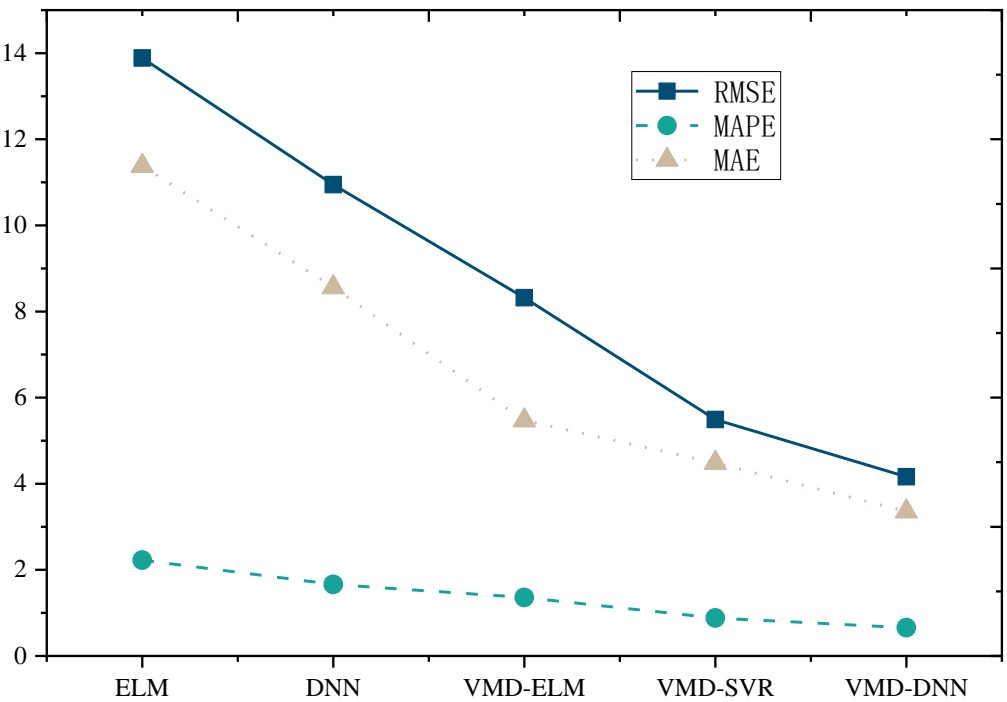

**Figure 6** Model comparison under online retail dataset.

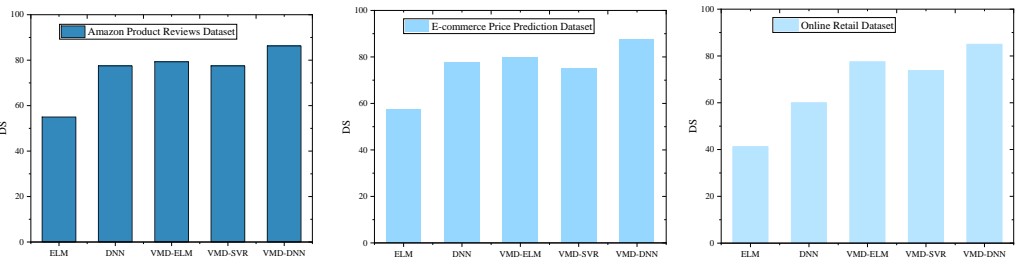

**Figure 7** DS comparison under different models.

these datasets, maintains its competitive edge by showing reliable directional symmetry, although it remains slightly inferior to the VMD-DNN model.

In comparison, the DNN and VMD-SVR models exhibit similar DS scores across the datasets but are notably lower than those of the VMD-DNN and VMD-ELM models. The lower DS scores for these models suggest that while they exhibit some level of directional accuracy, there remains substantial room for improvement in capturing directional trends effectively.

The VMD-DNN model exhibits notable advantages in both directional symmetry and prediction performance, particularly in the task of e-commerce product price prediction. This robust performance enhances the model's explanatory and practical utility. The VMD-ELM model, with its stability and commendable directional symmetry across

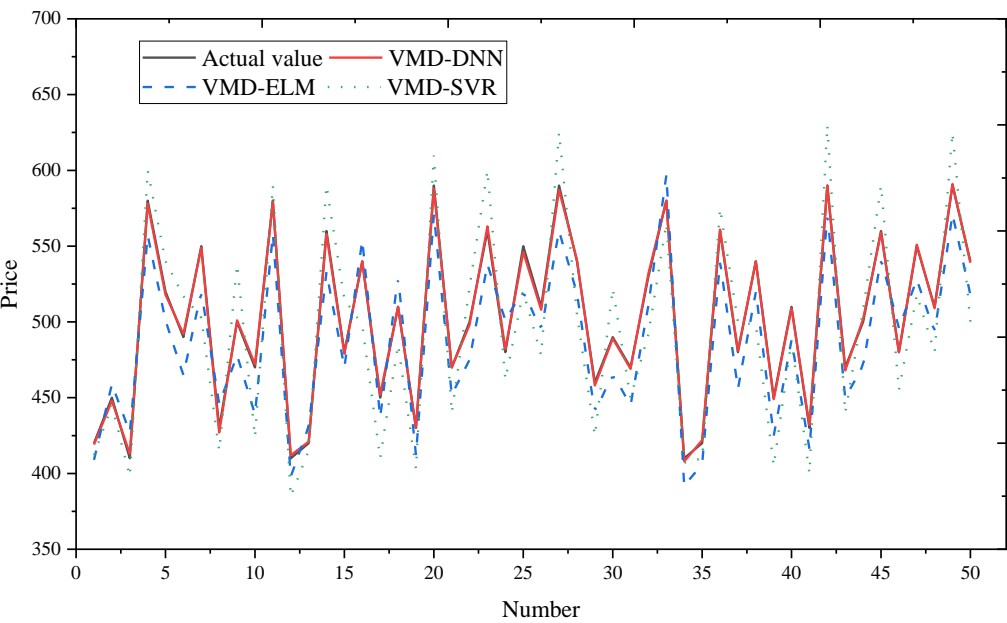

**Figure 8  Comparison of price forecast results for products in the 400–600 price range.**

datasets, emerges as a reliable alternative. The in-depth analyses presented here offer a comprehensive guide for selecting suitable prediction models, especially in scenarios where directional symmetry and interpretability are pivotal. Notably, the VMD-DNN and VMD-ELM models demonstrate strong performance in such scenarios.

To obtain final forecasts, the IMF decomposition results are linearly summed. Price forecast curves for e-commerce products in different price ranges are illustrated in Fig. 8 (400–600) and Fig. 9 (20–40), with actual observed values depicted in black for reference.

From the figures, it is evident that the predictions from VMD-ELM and VMD-SVR exhibit instability and the poorest alignment with the actual observations. In contrast, VMD-DNN demonstrates closer proximity to the actual observations than the single model. With the exception of occasional bias in the high-frequency subsequence in extreme cases, the predicted values for the remaining subsequences exhibit high compatibility with the actual values. Notably, the predicted trends closely mirror the actual trends.

The application of VMD to decompose the original e-commerce product price data, particularly within different price intervals, proves beneficial. This process effectively extracts and processes price fluctuation information, significantly enhancing the prediction performance of the DNN model.

## DISCUSSION

The application of the proposed VMD-DNN model in e-commerce product price prediction holds significant academic and practical implications. Firstly, the utilization of deep learning models can substantially enhance the accuracy of price prediction due to their potent nonlinear modeling capabilities. Traditional linear models struggle to capture the

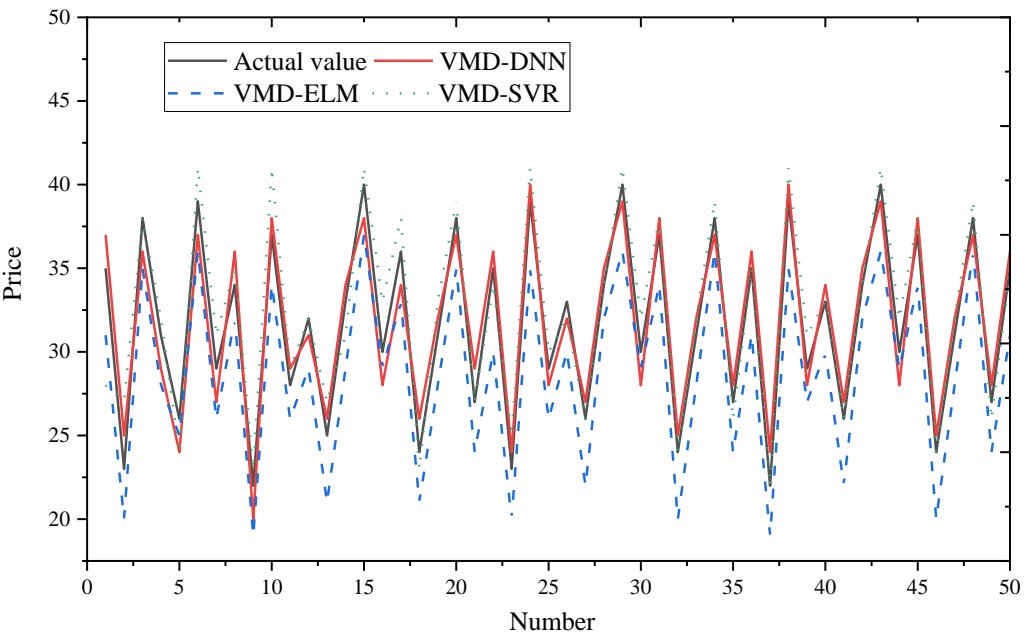

**Figure 9** Comparison of price forecast results for products in the price range 20–40.

intricate multi-level relationships underlying product prices. In contrast, deep learning, by learning patterns from extensive datasets, can better comprehend the influence of market factors, competitive dynamics, and other complex variables on product prices.

Secondly, the e-commerce marketplace's rapid changes and volatility underscore the importance of immediate and accurate price predictions. Deep learning models, being adaptable, can swiftly adjust predictions in response to seasonal market changes, promotions, and competitive pressures. This adaptability provides e-commerce platforms with more real-time decision support. Based on the experimental results, the proposed model plays a pivotal role in optimizing pricing strategies. A deeper understanding of product price trends and fluctuations enables e-commerce firms to formulate more effective pricing strategies, maximizing revenue and increasing sales. This balance between profit and market share in a competitive environment establishes a sustainable competitive advantage for businesses.

Furthermore, in terms of inventory management, accurate price forecasting aids in avoiding excessive or insufficient inventory levels. With the support of deep learning models, e-commerce platforms can plan inventory more precisely, reducing costs and increasing turnover, thereby improving overall operational efficiency.

However, the prediction challenges in e-commerce data arise from the influence of intricate factors like market competition, promotional activities, and user behaviors. These factors contribute to the nonlinear and nonstationary characteristics of the data, elevating the complexity of prediction tasks. Poor data quality or missing values can further pose challenges for VMD-DNN, as the model places high demands on data quality, potentially impacting its performance. Moreover, the variability in price fluctuations within the

e-commerce domain, influenced by seasonality, commodity characteristics, and other factors, may limit the generalization capability of VMD-DNN to specific scenarios. The model's performance may excel in certain commodities or time periods but may not be as effective in other cases.

Additionally, the utilization of complex VMD-DNN models may be excessive for certain straightforward prediction problems, where simplified models might offer more practical solutions. Lastly, the performance of VMD-DNN can vary across different types of time series data and prediction tasks. Validating the model's generalizability and adaptability across multiple domains and datasets is crucial to ensure its robustness in diverse contexts.

## CONCLUSION

After normalizing the data and applying the Variational Mode Decomposition (VMD) method, the optimal $K$ value is determined using the minimum fuzzy entropy criterion. Subsequently, a deep neural network (DNN) is employed to predict the modal components individually, with the predictions aggregated to produce the forecast values of the VMD-DNN fusion model. This methodology yields highly satisfactory fitting results for both the training and test datasets. Comparative analysis with other decomposition-integrated prediction models demonstrates that VMD-DNN exhibits significantly smaller prediction errors. This result underscores the model's effectiveness in decomposing complex e-commerce product price sequences into simpler sub-sequences, thus reducing the prediction burden on a single model and enhancing its generalization ability. The improved generalization ability contributes to the overall enhancement of prediction performance. The price prediction results not only provide valuable business insights but also serve as a scientific foundation for e-commerce executives to make informed decisions. The comprehensive analysis of extensive data allows business leaders to develop strategies that are more responsive to dynamic market changes. Future research could investigate the incorporation of dynamic features into time series data. Since e-commerce product prices are influenced by factors such as seasonality and promotional activities, integrating time dynamics into the model—through mechanisms such as time lag terms or periodicity models—could further enhance the model's ability to accurately capture the nuances of price series dynamics.

## ACKNOWLEDGEMENTS

I thank the anonymous reviewers whose comments and suggestions helped to improve the manuscript.

### Funding
The authors received no funding for this work.

### Competing Interests
The authors declare there are no competing interests.

## Author Contributions

- Haojie Wu conceived and designed the experiments, performed the experiments, analyzed the data, performed the computation work, prepared figures and/or tables, authored or reviewed drafts of the article, and approved the final draft.

## Data Availability

The code is available in the Supplementary File.

The dataset is available at Zenodo: Criado Moreno, H. P., & Del Pino Deniz Pérez, V. (2022). Lidl_prices [Data set]. Zenodo. https://doi.org/10.5281/zenodo.6430601.

## Supplemental Information

Supplemental information for this article can be found online at http://dx.doi.org/10.7717/peerj-cs.2353#supplemental-information.

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
