# Peer review of "Predicting e-commerce product prices through the integration of variational mode decomposition and deep neural networks"

_PeerJ Computer Science, doi:10.7717/peerj-cs.2353_

## Round 0.1 · original submission · Major Revisions

Please see both reviewer's comments. Reviewers suggest simplifying sentences, avoiding jargon, stating the problem and objectives early, providing overviews of VMD and DNN, ensuring a logical progression of ideas, and using clear headings and labeled figures. Additionally, it highlights the need to explain the significance of results, emphasize key findings, elaborate on the minimum fuzzy entropy criterion, and provide details on datasets, the DNN architecture, and training processes.

·

Basic reporting

• It covers most of the available data but many references relevent to this review are missing.
• At the end of the introduction, the goal and tasks for achieving it must be clearly defined. And in the conclusions, give a numbered list in accordance with the solved tasks
• Authors should check the MS for grammatical errors.

Experimental design

OK

Validity of the findings

OK

Reviewer 2 ·

Basic reporting

This is a meaningful study, by incorporating these suggestions, the deep learning network can be fine-tuned to achieve higher accuracy and better generalization, ultimately enhancing the e-commerce product price prediction model's performance.

Experimental design

- Incorporate advanced architectural designs such as Residual Networks (ResNet) or Long Short-Term Memory (LSTM) networks to capture temporal dependencies more effectively.
- Use more hidden layers and neurons to increase the model's capacity, ensuring that dropout layers are appropriately placed to mitigate overfitting.
- Implement learning rate schedules or adaptive learning rate optimizers like Adam to dynamically adjust the learning rate during training, enhancing convergence speed and model performance.
- Introduce regularization techniques such as L2 regularization to prevent overfitting and improve generalization capabilities
- Utilize normalization or standardization techniques to scale the input features, ensuring uniformity across the dataset.
- Conduct an extensive hyperparameter tuning using grid search or randomized search to identify the optimal set of hyperparameters for the VMD-DNN model.
- Experiment with different batch sizes, learning rates, and activation functions to determine the most effective combination.

Validity of the findings

- Consider additional evaluation metrics such as Mean Squared Error (MSE) and R-squared (R^2) to provide a more comprehensive assessment of model performance.
- Use cross-validation techniques to ensure the robustness of the model's performance across different subsets of data.

Reviewer 3 ·

Basic reporting

There are some minor basic structure and format issues authors should address:
Simplify complex sentences to make the text more accessible.
Avoid unnecessary jargon to ensure readability for a broader audience.
Clearly state the problem and the objective of the study early on.
Consider providing a brief overview of VMD and DNN for readers unfamiliar with these concepts.
Ensure a logical progression of ideas. For example, start with the problem statement, followed by the methodology, results, and conclusion.
Use headings or subheadings to organize sections if appropriate.
Ensure all figures (e.g., Figure 1, Figure 2, Figure 3) are clearly labeled and referenced

Experimental design

Provide more context about the datasets used for evaluation.
Please re-write the algorithm1 in more professional way

Validity of the findings

Explain the significance of the results, such as why the error reduction rates and DS scores are noteworthy.
Emphasize the key findings and their implications more clearly
Elaborate on the minimum fuzzy entropy criterion for selecting the optimal modal number (K).
Include details on the architecture and training process of the DNN used for forecasting.

---

## Round 0.2 · accepted · Accept

All reviewers have confirmed that the authors have addressed all of their comments.

·

Basic reporting

Accepted

Experimental design

ok

Validity of the findings

ok

Additional comments

no

Reviewer 2 ·

Basic reporting

The suggestions in the previous review have been incorporated by the authors

Experimental design

The suggestions in previous review have been incorporated by the authors

Validity of the findings

The suggestions in previous review have been incorporated by the authors

Reviewer 3 ·

Basic reporting

The authors have addressed all the previously pointed issues successfully.

Experimental design

The authors have addressed all the previously pointed issues successfully

Validity of the findings

The authors have addressed all the previously pointed issues successfully